# Injury Pain in Track and Field Athletes: Cross-Sectional Study of Mediating Factors

**DOI:** 10.3390/sports7050110

**Published:** 2019-05-10

**Authors:** Toomas Timpka, Jenny Jacobsson, Victor Bargoria, Örjan Dahlström

**Affiliations:** 1Athletics Research Center, Department of Medical and Health Sciences, Linköping University, 58183 Linköping, Sweden; jenny.jacobsson@liu.se (J.J.); bargoriavictor@gmail.com (V.B.); orjan.dahlstrom@liu.se (Ö.D.); 2Department of Orthopaedics and Rehabilitation, Moi University, Eldoret 30107, Kenya; 3Department of Behavioural Sciences and Learning, Linköping University, 58183 Linköping, Sweden

**Keywords:** sports medicine, pain perception, athletics, overuse injury, body consciousness, mediation analysis

## Abstract

This exploratory study aimed to investigate whether data from track and field athletes support a conceptual model where the perception of pain affects self-evaluated health directly, yet is also indirectly mediated through personal factors and external stressors. A cross-sectional study was performed among athletes (*n* = 278) competing at the highest national level in the sport. Data for the dependent and independent variables were collected using the EuroQol-5 dimensions (EQ-5D) standardized measure of health status, and the mediator variables were collected by the Body Consciousness Scale (BCS-HA), Perceived Motivational Climate in Sport Questionnaire (PMCSQ), Commitment to Exercise Scale (CtES), and Brief Cope survey instruments. Two hundred and twenty-five (81%) athletes provided complete data sets. A strong direct association (*c* = −16.49; *p* < 0.001) was found between pain and self-estimated health, and a global mediation was found through the mediators (mediation effect (ME), −1.814, 90% confidence interval (CI), −3.414, −0.351). Specific mediation was observed for body competence (ME, −0.721; 90% CI, −2.028, −0.097) and private body consciousness (ME, −0.514; 90% CI, −1.761, −0.017). In conclusion, we found a negative association between perceived pain and self-evaluated health among competitive track and field athletes, and we found that a portion of the association was mediated through mediating factors. Under the assumption that the analyzed pain data primarily represented noxious pain, the conceptual model of the relationships between pain and self-estimated health among high-level track and field athletes was supported.

## 1. Introduction

During the past two decades, injury rates and patterns have been established in track and field; studies monitoring athletes over entire seasons have showed alarmingly high injury incidence proportions and have showed that a large majority of the injury episodes are associated with sports overuse [1,2]. Overuse injuries are characterized by gradual onset and causal mechanisms associated with inadequate rest following bouts of exercise [3,4]. In a recently developed stepwise model of injury risk among track and field athletes, a training load index was found to be replaced by maladaptive coping behavior [5]. Consequently, the timing of the exercise load appears to be more important in injury causation than the relative quantity of the load. This observation suggests that psychological and social factors that disturb adequate interpretations of pain and discomfort and consequential changes in exercise load are important when investigating mechanisms causing overuse injury in track and field athletes.

In sports that require extensive capabilities with regard to speed, endurance, or power, athletes use body perceptions to regulate exercise load. Primarily, the athletes need these perceptions to manipulate power output over exercise bouts in order to balance energy expenditure and speed in a way that will allow an optimal completion of the activity [6,7]. In addition, competitive athletes must have the ability to interpret and adjust the level of pain when pushing physical capacity during bouts of exercise [8]. With a balanced increase in pain tolerance, in association with functional over-reaching, the athlete can improve physical performance. Track and field athletes competing at a high level must therefore, on a day-to-day basis, distinguish ordinary muscle soreness pain resulting from functional over-reaching from noxious pain that constitutes a warning sign of overuse injury. The separate measurement of these pain types is methodologically challenging. However, the total pain that an athlete is prepared to accept for a particular exercise has been reported to be affected by a variety of personal factors, ranging from internal motivation to previous experience [9,10]. In addition, the pain tolerance among competitive athletes may be affected by external social stressors, e.g., the perceived level of support from coaches and fellow athletes [11]. 

The aim of this exploratory study is to investigate whether data from track and field athletes support a conceptual model where the perception of pain affects self-evaluated health and training effects both directly and indirectly, i.e., is also mediated through a set of personal factors and external stressors. These mediators include body consciousness, coping strategies, rigidity in exercise behaviors, and motivational climate. The analytic model is based on the assumption that muscle soreness pain caused by functional over-reaching during training is not considered harmful and does not impact self-estimated health among track and field athletes.

## 2. Materials and Methods

A cross-sectional study design was used. The research population consisted of Swedish male and female elite track and field athletes. Individuals ranked among the national youth and adult top ten in each track and field event were asked to participate in a 1-year surveillance study, regardless of their injury status. The 278 athletes participating in this surveillance study were also invited to respond to a psychological survey.

Data for the psychological survey were collected from the athletes during the pre-competition period. The data on pain, self-reported general health, and the psychological mediator variables were collected by a postal survey. In parallel, a web questionnaire was distributed to ask for personal sport-specific and sociodemographic data (SiteVision V.2.5, Senselogic AB, Örebro, Sweden).

A conceptual model was developed (Figure 1) where sports injury and functional over-reaching are assumed to be causes of symptoms in terms of pain, noxious as well as functional. Symptoms are thereafter effecting, both directly and indirectly (by mediations) through personal and external factors, outcomes in terms of health and training effect. While pain or discomfort associated with injury causes a decrease in self-estimated health among track and field athletes, it is assumed that the effect of pain associated with functional over-reaching on self-evaluated health is minimal.

The effect of pain on perceived health and training effect is assumed to be direct, but also indirect by the mediating factors of personal or external (social) character. Personal factors that are assumed to be associated with pain perception, and in turn will effect outcomes, are body consciousness, hyperactivity, coping strategies, and rigidity in exercise behaviors. An example of an external social factor is the perceived motivational climate [12].

Due to the fact that the survey asked about pain in general (and not noxious and functional pain separately), and since functional pain is an inherent component of everyday life as part of being a track and field athlete, we assumed that this general measure of pain only represented noxious pain to a certain degree. According to the conceptual model, noxious of pain is related directly and indirectly to (self-reported) health. The part of the conceptual model being tested is presented in Figure 2.

The EuroQol-5 dimensions (EQ-5D) instrument [13] was used to collect data for the main study variables. This instrument collects self-reported data on mobility, self-care, usual activities, pain and discomfort, and anxiety or depression. The respondents are also asked to estimate their general health on a graphical thermometer calibrated from 0 (“worst imaginable health status”) to 100 (“best imaginable health status”). In this study, only the sections on pain and discomfort and general health status were used. The athletes estimated their health on a visual analog scale graded from 0 to 100, with 0 being the "worst state" and 100 "the best possible state of health”; pain and discomfort was measured using a 3-graded scale (no pain or discomfort, moderate pain or discomfort, extreme pain or discomfort).

### Mediator Variables

Body consciousness and hyperactivity. Body consciousness ratings were founded on the Body Consciousness Scale (BCS) [14]. To integrate aspects of hyperactivity and attention deficiency, the BCS-HA questionnaire (21 items; the last six items addressed hyperactivity and attention deficiency) was constructed by adding items from the hyperactivity definition in DSM-IV. Scores were recorded and computed separately for private body consciousness, public body consciousness, body competence, and hyperactivity.Coping strategies. The strategies employed by the athletes to understand and manage perceptions of pain were recorded using the Brief Cope instrument [15]. This instrument covers both adaptive strategies such as active coping, emotional support, instrumental support, positive reframing, planning, mood, acceptance, and religion, and maladaptive strategies such as self-distraction, denial, substance use, behavioral disengagement, venting, and self-blame.Rigidity in exercise behaviors. Tendencies for over-commitment to training were estimated by the Commitment to Exercise Scale (CtES) [16]. The CtES is an eight-item questionnaire designed to assess an individual’s psychological commitment to the activity of exercising. Perceived motivational climate. The pattern of demands in the athlete’s local sports setting was recorded using the Perceived Motivational Climate in Sport Questionnaire (PMCSQ). The PMCSQ includes separate scores for mastery/task accomplishment and performance orientations [12].

All data were first presented using descriptive statistics, i.e., means and standard deviations for continuous data and frequencies for categorical data. Differences in the distribution of non-responders and pain prevalence were analyzed using Chi-square tests. The direct association between pain and health was examined in a basic simple model, and thereafter in two models with sex and age group used as moderators (i.e., testing if the association between pain and health was different over different levels of either sex or age). Thereafter, to examine the relationship between pain and health while mediated by personal and external factors, models where tested both with each mediator separately and with all of the mediators together in a multiple mediation model (Figure 2) [17,18]. Each specific mediation is represented by the multiplication of the relation between Pain and the mediator (*a*_i_) and the relation between the mediator and Self-estimated health (*b*_i_), i.e., *a*_i_*b*_i_, and in the multiple mediation model the total mediation is the sum of those specific mediations.

Due to the fact that the independent variable of perceived pain was not explicitly measuring noxious pain, and the fact that an exploratory approach was used to examine if a model fitted the data, *p* < 0.10 was regarded as statistically significant. Accordingly, 90 percent confidence intervals (CIs) were estimated using the PROCESS macro for Statistical Package for the Social Sciences (SPSS) version 23.0 (IBM SPSS Statistics, New York, USA).

The study design was approved by the Research Ethics Board in Linköping (dnr. M-201–08). Informed written consent was obtained from all participants in the study before the baseline surveys were distributed. For those under 18 years of age, approval was also obtained from their parents. The funding organization had no influence on the collection of data, their analysis and interpretation, nor the right to approve or disapprove the publication of the finished manuscript [19].

## 3. Results

Two hundred and twenty-five (81%) of the invited athletes provided complete data sets; 42% (*n* = 95) of the respondents were female (Table 1). There was no statistically significant difference in the distribution of non-response with regard to age, sex, or event group. The mean age of the survey respondents in the adult category was 24 years (range, 18–37 years) and the mean age of the youth respondents was 17 years. Sixty-two percent (*n* = 140) of the athletes reported no pain at the time of the data collection, and 38% reported moderate pain. No athlete reported having severe pain. Pain was reported by 36% of the boys, 25% of the girls, 48% of the men, and 36% of the women; there was no significant difference between the athlete categories with regard to pain prevalence (*p* = 0.12).

### 3.1. Direct Association between Pain and Self-Estimated General Health

As assumed in the analytic model, a strong direct association was observed between total (reported) pain and self-estimated general health (*c* = −16.49; *p* < 0.001). Neither sex (*p* = 0.255) nor age group (*p* = 0.362) qualified as factors moderating the association between pain and self-estimated general health.

### 3.2. Mediated Associations between Pain and General Health

In the multiple mediation model, a global-level mediation of the association between pain and self-estimated health through the mediator variables was observed (mediation effect (ME): −1.814; 90% CI, −3.414, −0.351) (Table 2). A specific mediation between pain and health was observed for body competence (ME: −0.721; 90% CI, −2.028, −0.097) and private body consciousness (ME: −0.514; 90% CI, −1.761, −0.017). These factors mediated a negative association between pain and self-reported general health. No statistically significant mediation between pain and general health was observed for motivational climate, coping strategies, or commitment to exercise. In addition, no moderating effect was observed on the mediation with regard to sex or age group.

## 4. Discussion

This study set out to investigate whether data from track and field athletes support a conceptual model where the perception of pain affects self-evaluated health, both directly and indirectly mediated through a set of personal factors and external stressors. We found, as expected, a negative association between perceived pain and self-evaluated health. Further, we observed a global-level mediation of this negative association through the mediator variables. In other words, pain reduced the self-evaluated health among the athletes, and the reduction was reinforced through the mediating factors. A high tolerance of pain has commonly been found among sportspersons [20], as well as a negative correlation between conditioned pain modulation (CPM) and fear of pain [21]. On the other hand, non-athletes with long-term pain have been found to exhibit poor CPM compared with controls [22,23]. Although both athletes and non-athletes may experience pain on a daily basis, the effect on pain modulation seems to be opposed. However, it is not evident that this difference is only associated with a background in sports. Track and field athletes may experience anxiety when they perceive pain that they associate with an injury, similarly to when non-athletes with previous long-term pain experience pain catastrophizing when encountering a new episode [24]. Nevertheless, pain associated with functional over-reaching seldom evokes fear among athletes, but rather brings to mind feelings of satisfaction. Thus, injured athletes may experience negative psychological influences from certain types of pain manifesting as worry and mood changes [25], whereas non-injured athletes may experience positive influences from other types of pain, e.g., such pain construed as muscle soreness (“I have trained well”) [26,27]. This implies that whereas noxious injury pain and muscle soreness pain may evoke comparable bottom-up signals, their higher order evaluations may evoke top-down influences that either diminish or improve pain modulation. Such an interpretation is underpinned by the fact that electroencephalography and functional magnetic resonance imaging responses to nociceptive stimuli have been shown to originate from an extensive network of brain regions that is considered to reflect neural activities mediating pain experiences [28,29]. The strong negative association between perceived pain and self-evaluated health, directly and indirectly via the mediator variables, found in this study supports the assumptions of the conceptual model, including the assumption that the athletes’ reported data represented noxious type pain. 

In this study, we thus found a global-level mediation of the association between pain and self-estimated health through the mediator variables, and indications of a specific mediation between pain and health for body competence and private body consciousness. Body awareness is of fundamental importance for most athletes, yet it is often taken for granted. Recent research has outlined the cortical body matrix model [30] for detailed studies of pain experience and performance of complex movements [31]. Whereas private body consciousness connotes the athlete’s disposition to focus on internal bodily sensations, body competence reflects skills related to the evaluation of effective body functioning [14]. However, the mechanisms behind the observed mediated effects cannot be considered as determined. For instance, it is possible that the effects are a consequence of long-term exposure to overuse injury pain. In track and field athletes, overuse injury leads to athletes suffering noxious injury pain for lengthy periods of time, lasting from weeks to months [2]. In non-athletes, long-term pain has been associated with disturbances in body perception in which the affected body part is neglected or misperceived. Among patients with complex regional pain syndrome (CRPS), a majority have been reported to have at least one such neglect-like symptom [32]. In other long-term pain settings, sensory abnormalities related to the affected body part have been reported, e.g., unawareness of limb position [33]. Several forms of long-term pain, including chronic back pain [34] and CRPS [35], have also been found to be associated with distortions of the perceived size or shape of the affected body part. Thus, the mechanisms behind the mediation effect observed in this study warrants further investigation.

This study has methodological strengths and limitations that need to be taken into account when interpreting the results. A strength of the study is that it is one of the first investigations to use an analytic model to distinguish between perceptions of injury-related and non-injury-related pain in athletes. Distinguishing between different types of pain is essential for athletes and poses challenges for researchers. Pain was measured using the EQ-5D instrument, which has been shown to be valid for the epidemiological recording of pain and is frequently used in studies of quality of life among patients with long-term pain [36]. Further studies should include specific measurements constructed to distinguish between noxious and functional pain. A limitation of the study is that a cross-sectional design was used, which restricts the possibility of drawing inferences about causation. It has been proposed that mediation analyses should be constrained to settings where the postulated time-ordered relationship of variables is defensible [37,38]. We acknowledge that the specific time ordering of variables is largely underemphasized in applications of mediation analysis [39]. However, two critical assumptions underpinning the theoretical model used in the present analyses were that the perception of noxious pain precedes a decrease in self-evaluated health, and that this association is mediated through factors which are effected before the decrease in self-evaluated health. While acknowledging the above-mentioned circumstances regarding pain and body consciousness, we infer that these two assumptions have not been violated in the analyses. Nonetheless, based on the exploratory nature of the study design, it is highly warranted that the analytic model is applied in longitudinal studies to validate the results.

In conclusion, we found a negative association between perceived pain and self-evaluated health among competitive track and field athletes, and we found that a portion of the association was mediated through mediating factors. Under the assumption that the analyzed pain data primarily represented noxious pain, the conceptual model of the relationships between pain and self-estimated health among high-level track and field athletes was supported. These results warrant confirmation in studies using prospective designs.

## Figures and Tables

**Figure 1 sports-07-00110-f001:**
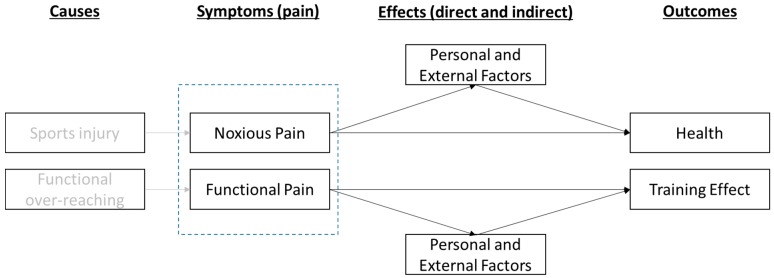
Conceptual model of the causes of pain and how health and training outcomes are effected directly and indirectly through mediation by personal and external factors.

**Figure 2 sports-07-00110-f002:**
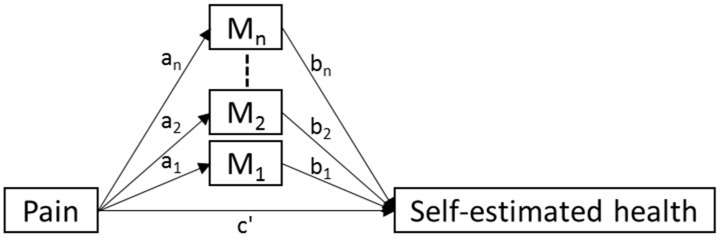
Simplified overview of the basic analysis model used for the study, where M_1_, M_2_, … M_n_ represents the personal and external mediators.

**Table 1 sports-07-00110-t001:** Overview of survey response.

Study Measures	Boys *n* = 61	Girls *n* = 40	Men *n* = 69	Women *n* = 55
**-**	Mean (SD)	Mean (SD)	Mean (SD)	Mean (SD)
Self-estimated health (EQ-5D)	76.56 (21.62)	83.70 (14.17)	78.99 (15.99)	84.18 (13.21)
Motivational climate (PMCSQ)				
Mastery	1.86 (0.57)	1.80 (0.64)	1.80 (0.43)	1.90 (0.51)
Performance	3.08 (0.66)	2.87 (0.63)	3.05 (0.69)	2.85 (0.51)
Commitment to exercise (CtES)				
Rigidity in exercise behavior	85.43 (13.81)	84.00 (13.91)	83.77 (14.34)	82.23 (13.15)
Body consciousness (BCS-HA)				
Private body consciousness	57.48 (28.60)	56.83 (22.65)	55.14 (23.22)	56.54 (27.42)
Public body consciousness	38.61 (21.13)	52.89 (26.26)	52.58 (25.99)	65.10 (26.37)
Body competence	45.88 (27.37)	44.13 (23.74)	56.21 (30.66)	46.57 (30.86)
Hyperactivity	62.75 (16.32)	63.58 (18.48)	60.69 (17.15)	59.57 (15.72)
Coping strategies (Brief Cope)				
Adaption	2.59 (0.39)	2.50 (0.44)	2.65 (0.41)	2.66 (0.37)
Maladaption	1.92 (0.36)	1.85 (0.42)	1.87 (0.34)	1.82 (0.32)

**Table 2 sports-07-00110-t002:** Mediation of the association between pain and self-reported general health in elite track and field athletes (*n* = 225; 90% confidence interval (CI)).

Mediators	Mediation Effect	90% CI
Total mediation		
All factors	−1.814	−3.414, −0.351
Body consciousness (BCS-HA)		
Private body consciousness	−0.514	−1.761, −0.017
Public body consciousness	0.261	−0.137, 1.172
Body competence	−0.721	−2.028, −0.097
Hyperactivity	−0.078	−0.742, 0.087
Motivational climate (PMCSQ)		
Mastery	−0.202	−0.967, 0.038
Performance	−0.035	−0.527, 0.111
Commitment to exercise (CtES)		
Rigidity in exercise behavior	0.015	−0.176, 0.458
Coping strategies (Brief Cope)		
Adaptive	−0.314	−1.131, 0.018
Maladaptive	−0.226	−1.185, 0.082

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
