# Peer review of "Injury Pain in Track and Field Athletes: Cross-Sectional Study of Mediating Factors"

_sports, 2019, doi:10.3390/sports7050110_

Round 1

Reviewer 1 Report

General Comment

The aim of the manuscript sports-436166 was to investigate whether perception of noxious injury pain among track and field athletes is affected by body consciousness, coping strategies, motivational climate, and commitment to exercise. This is just a cross-sectional, exploration study, thus several limitations are present here. The paper needs extensive corrections. As for example, the literature review but mostly the rational is very weak, while the introduction section is not linked with the research question or the aim of the present study. In addition, due to the fact that authors performed this study in top athletes, with regular, high loaded training regiments, they should analyze which is the interplay between physiological demands, during a race or during the training sessions, and the psychological stress. Another important issue in this study, is that even if it explores the functional over-reaching and the link with psychological stresses, nowhere in the manuscript is described when this study has been performed. They conduct this study during which period of the training session? In many cases, authors implies that  the cause of overuse injury, is the pain and discomfort and etc. This is totally not true. Overuse injuries are outcomes of the extensive physiological stress that extensive training loads combined with adapque rests intervals induce. Thus authors should revise their rational. According to these comment as well as the analytical comments below, this manuscript has significant faults, errors and limitations, which does not allow it to be consider for publication as it is. Authors should follow the instruction of the journal for manuscript format.

Specific comments.

Introduction

A general comment for this section.

Introduction is very weak, with a not solid rational. Authors should rewrite the introduction section, taking into consideration to discuss analytically, the interplay between physiological demands, during a race or during the training sessions, and the psychological stress. As for example, authors should provide a brief report about the mechanism that link extensive training loads with the psychological terms that they want to explore. A more detailed description about the reports of the literature in the field is also needed. Authors should take into consideration, that this paper will be read by non expert also. Thus, they have to provide more details about the body consciousness, motivational climate, coping strategies, and commitment to exercise. Finally, the pain and the muscle soreness are outcomes of the training type, frequency, load and the ration between training and rest intervals. However, this is not discussed anywhere.

Page 1 Lines 34-37. This sentence provides a very important message. Thus authors should analyze in a more extended matter the results of this study, while they have to describe the physiological base of this report.

Page 1 line 41. Please rephrase the following sentence: “exercise load based on Body perceptions are central.”

Page 1 Lines 41-43. This is definitely not the description of the term “pace”. Please rephrase accordingly

Page 1-2 Lines 43-45. Authors if decide to use this terms, should describe better, for which track and field athletes this is true. In addition, they must, be more specific in entire introduction about which athletes they are talking about

Page 2 Lines 50. Please analyze these factors

Page 2 Lines 53-57. Why authors do this study. As the introduction section is right now, there is not a logical link between the rational of the introduction and the aim of the present study. Are they any hypotheses in this study? Where the assumption that authors uses for their model based on?

Materials and Methods

When this study conducted? The training period that the data were collected is of highly importance.

Page 2 lines 60-68. In this sentence authors implies that the cause of overuse injury, is the pain and discomfort? This is totally not true. Overuse injuries are outcomes of the extensive physiological stress that extensive training loads leads. Based on which reports this model was development? If this model was not based to previous studies, authors should provide detailed description about the methodology as well as the rational which leads them to this model. In addition, provide refs or the base for the assumption that described in this paragraph.

Page 4 lines 81-84. A more detailed description about the participants of the present study is of highly importance. Authors should provide a detailed description about the number of participants in each track and field events, the training background, training frequency, loads and etc, as well as their age, sex, anthropometrics characteristics, athletes rank and etc. As it is right now, the description of the participants is not enough, and thus the reader does not know, how, when or to whom the outcomes of the present study will be used. Finally, authors should provide a detailed description about the training season that this study was conducted. They collect their data during Preparatory, Pre-competition, Competition or Transition season? And all the data of all participants were collected in the same period? In addition, authors should provide data about the injuries that their participants have during the study.

The questionnaires that they used have been already used in other studies or they have made them. Please provide refs as well as the reliably and validity data.

Page 3 lines 117-120. Please provide data about the methodology that you used for the determination of the two models

Why authors used p<0.10, while for such type of study p has to be lower than 0.05?

Please provide the power analyses results, for the strength of your study. 225 participants are enough for such type of study?

Results

Detailed description about the mean and the sd in each value and the p values should be reported.

Paragraph 3.1. Please analyze

Including in the same models, men, women, boys and girls is considered right? Authors should explore the possible models, in each category separately.

Authors should consider to use either linear regression analyses or something like this, to estimate the specific contribution of each mediation

Discussion

Authors should re-write the discussion section according to the results of the suggested analyses

What was the main finding(s) of the present study?

Please discuss your findings in a more physiological manner.

Author Response

Please see separate file.

Reviewer 2 Report

This paper reports a self-estimated injury report based on EQ-5D survey and relating the cause and effect through mediator using the BCS-HA, PMCSQ, CtES, and 17 Brief Cope instruments. This paper is well written; however, the type of questionnaire will dictate the results as far as I can think, I wonder if the authors used the EQ-5D survey identical questions to what extend they have alter them. The EQ-5D has also 3 different survey and which one they have used? A visual representation of the causal variable and the outcome. The mediation is similar to the multiple regression analysis which the ability to infer the cause and effect. Therefore, I suggest a graphical representation is a plus.

I asked the above questions for clarity, however, the authors did not include any explanation of why they don't use any graphical representation? 

You could have highlighted what changes you have done!

please revise the writing in the first paragraph of the introduction, it is written poorly and is long. 

Author Response

Please see separate file.

Reviewer 3 Report

Overall, this is a good and novel concept that will be of interest to practitioners.  It shines a light for some into an area that isn't well understood or even thought of for most S&C practitioners in the areas of body competence and body consciousness.  For the most part it was a straight forward manuscript.  

I do have a few areas for clarification

Why was the alpha priori set for 0.10 ?  I am guessing that this is due to the use of two-sided statistical tests, please explain this for the readership who may not be as statistically savvy.  

On page 7, line 226 and 227, you acknowledge the issues but infer that these assumptions have not been violated.  Could you expand on why this is?  This was something that I questioned as this was cross sectional rather than longitudinal.  

Please review the entire manuscript for grammatical and spelling errors

Author Response

Please see enclosed file.

This manuscript is a resubmission of an earlier submission. The following is a list of the peer review reports and author responses from that submission.

Round 1

Reviewer 1 Report

This paper reports a self-estimated injury report based on EQ-5D survey and relating the cause and effect through mediator using the BCS-HA, PMCSQ, CtES, and 17 Brief Cope instruments. This paper is well written; however, the type of questionnaire will dictate the results as far as I can think, I wonder if the authors used the EQ-5D survey identical questions to what extend they have alter them. The EQ-5D has also 3 different survey and which one they have used? A visual representation of the causal variable and the outcome. The mediation is similar to the multiple regression analysis which the ability to infer the cause and effect. Therefore, I suggest a graphical representation is a plus.

Minor Comments:

Page 3, line 101: The beginning of the section must be written bold, I am guessing, “Body consciousness and hyperactivity.”

Page 3, line 116: commitment to the exercise.

Reviewer 2 Report

Is the study on track and field athletes or a variety of athletes?  The use of the term athletics can either stand alone or track and field athletes can stand alone but both are not needed as one is a broad term for sport qualities while the other is specific to the participant in a sporting event.  Current format does not make sense and is not needed, pick one terminology and use throughout the manuscript.

Line 12.  The wording does not make sense.  Rewrite for clarity.  “…among Athletics (track and field) athletes varies with…”

Line 16-17 have acronyms absent of clarification.  Add the words they referring to

Remove Athletics throughout the manuscript and use track and field since this appears to be the population studied.

All citations need to come before the period throughout the manuscript.

The introduction is lacking discussion or at least recognizing other stressors that are contributors to overuse injuries such as, excessive travel, collegiate academics, social elements, etc.

There is no mention by the authors how these athletes were determined to have “overuse injuries”.  This absence needs to be addressed because if a self-reported overuse injury then this an assumption of participant truthfulness. 

The authors need to further explain what the specific variables were analyzed and why.

Example: Body consciousness is mentioned but why is this important and provide a definition (introduction) & example in the discussion section.

The discussion provides no connection of the results and its value to coaching application. How and why is a coach going to be interested in the information presented. 

The discussion section needs to be rewritten for improved flow as the points trying be made by the authors skip from concept to the next.

There are too many manuscript errors at this point to provide a line for line comments as it needs to be redone in its entirety for clarity.